# Identification of Experimental and Control Areas for CCTV Effectiveness Assessment—The Issue of Spatially Aggregated Data

**Adam Dąbrowski [1], Piotr Matczak [2,*] , Andrzej Wójtowicz [3] and Michael Leitner [4]**

[1]   Institute of Geoecology and Geoinformation, Faculty of Geographical and Geological Sciences, Adam Mickiewicz University in Poznań, Bogumiła Krygowskiego 10, 61-680 Poznań, Poland; adam.dabrowski@amu.edu.pl

[2]   Institute of Sociology, Faculty of Social Sciences, Adam Mickiewicz University in Poznań, Szamarzewskiego 89c, 60-568 Poznań, Poland

[3]   Faculty of Mathematics and Computer Science, Adam Mickiewicz University in Poznań, Umultowska 87, 61-614 Poznań, Poland; andre@amu.edu.pl

[4]   Department of Geography and Anthropology, Louisiana State University, E-104 Howe Russell Geoscience Complex, Baton Rouge, LA 70803, USA; mleitne@lsu.edu

*   Correspondence: matczak@amu.edu.pl

**Abstract:** Progress in surveillance technology has led to the development of Closed-Circuit Television (CCTV) systems in cities around the world. Cameras are considered instrumental in crime reduction, yet existing research does not unambiguously answer the question whether installing them affects the number of crimes committed. The quasi-experimental method usually applied to evaluate CCTV systems' effectiveness faces difficulties with data quantity and quality. Data quantity has a bearing on the number of crimes that can be conclusively inferred using the experimental procedure. Data quality affects the level of crime data aggregation. The lack of the exact location of a crime incident in the form of a street address or geographic coordinates hinders the selection procedure of experimental and control areas. In this paper we propose an innovative method of dealing with data limitations in a quasi-experimental study on the effectiveness of CCTV systems in Poland. As police data on crime incidents are geocoded onto a neighborhood or a street, we designed a method to overcome this drawback by applying similarity measures to time series and landscape metrics. The method makes it possible to determine experimental (test) and control areas which are necessary to conduct the study.

**Keywords:** crime analysis; urban landscape; GIS; landscape analysis; Closed Circuit Television (CCTV); Poland

## 1. Introduction

The progress in surveillance technology has led to the development of Closed-Circuit Television (CCTV) systems in cities around the world. However, studies on the impact of CCTV cameras on the number of crime incidents do not provide an unambiguous answer [1]. Practitioners often emphasize the impact CCTV cameras can have both in terms of the ability to identify a criminal and as a prevention method that deters criminals [2,3]. Yet, a review of CCTV impact on crime reduction research by Lim et al. [1] shows inconclusive findings. Although the effect is noticeable for vehicular crimes and parking lots, it is usually not significant for city centers and residential areas. The ambiguous evidence of the effect necessitates careful examination of theoretical assumptions concerning specific types of crime and the role of contextual variables. It implies tailoring adequate methods.

A thorough look at particular studies reveals several methodological problems with measuring CCTV camera impact on crime. The randomized experimental design is difficult to achieve in practice [4]. Another issue is that the design of valuation studies does not always satisfy the criteria of validity. Specifically, this may happen in simple before-and-after designs applied in a particular location [4,5]. The quasi-experimental method is the second best option available for assessing the impact of CCTV on crime prevention [4,6], after the randomized experiment [7]. A series of quasi-experimental studies have been carried out in the last 15 years. Several of them have focused on crime displacement issues related to the evaluation of CCTV cameras [1,3,8] and on measuring the extent of spatial crime displacement (see: WDQ Spreadsheet Calculator; [9]). These studies rely on designing experimental, control, and buffer areas located in direct proximity. However, Welsh and Farrington [10] suggest that experimental and control areas should not be adjacent (http://www.jratcliffe.net/software/wdq-spreadsheet-calculator/). This approach entails problems with the data in specifying proper experimental and control areas. In particular, the level of spatial data aggregation can hamper identification of adequate experimental and control areas. This issue has not been of interest so far in the study of CCTV effectiveness [11]. However, it has already been noted in geographic studies concerning, e.g., interventions on rivers [12] or in urban landscapes [13].

Quasi-experimental studies offer the most robust evaluative assessments. Relying on real-life situations (not designed by an evaluator), this method tests the effect of an intervention (installation of a camera) in two areas: The experimental (where the intervention occurs) and the control (without intervention). Having data on the number of crimes committed before the intervention and after, one can determine the effect. Nevertheless, the method faces difficulties. The main challenge lies in ensuring that both areas differ by the intervention only. If this condition is fulfilled, we can assume that the change is the effect of the intervention [14]. Thus, the selection of appropriate experimental and control areas is crucial [4]. As the method depends on position of the existing cameras, the challenge is to identify areas with available data allowing for the calculation of the effect. Current research reveals data deficiencies with regard to: (a) Data on crime, which are necessary for determining the effect; and (b) data on site characteristics, which are indispensable for ensuring that both areas are similar to the extent that they may be treated as comparative [15]. The quasi-experimental method is easy to apply if address-level data are available, as is the case in most Anglo-American countries. However, analyzing the efficiency of cameras is problematic in countries where available data from the police are not precise.

In this paper we focused on a specific example where data was problematic, i.e., where locations of crime incidents are not attributed to addresses but to streets. We proposed an innovative method of selecting experimental and control areas with this data limitation, based on time series and landscape analysis. The scope of this study was limited. We concentrated on the crime prevention effect of CCTV cameras, leaving aside the detection aspect. The method was designed to be used when crime incident records are aggregated to larger spatial units, such as streets or neighborhoods. It is based on similarity measures of time series and landscape metrics used to select experimental and control areas, consistent with conditions suggested in the literature for performing the effectiveness analysis. In this paper, the proposed method is applied to the city of Poznań, Poland.

## 2. Methodological Challenges in Evaluating Effectiveness of CCTV Using the Quasi-Experimental Method

Most studies on the influence of CCTV camera installation or other crime rate interventions rely on the quasi-experimental design [16]. This design requires choosing experimental and control areas to assess the impact of an intervention. An experimental area is usually defined as an area covered by one camera, which is interpreted in a simplified way as a buffer area of a given distance from the camera [4], or as a visibility area designated by Geographic Information System (GIS) tools [17]. Where cameras are placed densely and the areas they cover overlap, the areas are combined into one experimental area [3]. Experimental areas need to have at least one camera installed, and a mounted

camera should be the main intervention taking place in that experimental area. A control area should not be directly adjacent to the experimental area. This rule is violated when the concentric buffer areas approach is applied [4,11,16,18,19]. Moreover, researchers studying the effectiveness of CCTV systems suggest that there should be at least 20 crime incidents before the intervention [16], and the number of crime incidents recorded should cover at least one year before and one year after the cameras were installed [6]. Although the quasi-experimental method makes it possible to analyze effects of a policy intervention, the method involves several difficulties. First, data on crime incidents both in experimental and control areas are not always available [16]. Second, even if data are available, in some cases the number of crime incidents is small, which makes robust statistical conclusions impossible [1]. For particular experimental and control areas, the number of actual crime incidents needs to be sufficient to allow for the application of statistical tests. Since a pair of experimental/control areas must be as similar as possible in terms of crime rate and socio-economic characteristics, it may be challenging to meet this condition. A possible solution for this problem is to aggregate data spatially or temporally (e.g., by analyzing crime incidents over a larger area and/or over a longer period), or to aggregate different crime categories. Third, this approach leads to another problem. As the installation of a CCTV impacts different crime categories differently, careful decisions must be made on the selection of crime categories considered in a study. For instance, car robbery and car theft, which concern the same subject (cars) are treated as two different felonies. To detect CCTV impact, Welsh and Farrington [16] suggested to analyze crimes related to car theft and battering (street fights) only. They are relatively common crime types that occur due to a "window of opportunity" [20].

Fourth, although it is contested that the installation of CCTV cameras is the cause of the drop in crime [21], the crime rate has been dropping in European cities for more than a decade [22–24]. This drop coincides with the development and spread of CCTV technology, which makes it difficult to isolate the influence of CCTV cameras alone. Farrell et al. [25] examined 17 hypotheses explaining the drop in crime. For Poland, Siemaszko [26] proposes seven hypotheses, including: (a) A decrease in the age group inclined to commit a crime (as a result of an aging population), (b) emigration, (c) relative decrease of the value of goods desired by criminals, (d) better work by law enforcement, (e) more guards, (f) more street lights, and (g) use of a CCTV camera. These hypotheses were not systematically tested, however.

Fifth, selecting appropriate control areas is challenging since they must be similar to the experimental areas in both criminal activity and other attributes of areas to be considered "twin areas" [17,27]. Sixth, the level of spatial aggregation of crime incidents data can present challenges as well. Current methods for selecting experimental and control areas are based on buffers around cameras. Buffers are sufficient if crime incidents are attributed to exact addresses or geographical coordinates [1,27,28]. However, they cease to work if crimes are geocoded to bigger areas such as streets or neighborhoods. Research conducted on aggregated crime data necessitates that these data are disaggregated [29]. Alternatively, an analysis of the relationship between crime and spatial characteristics, such as unemployment [30] or land use [31,32], may be performed. Spatially aggregated crime data are problematic when conducting research on the effect of an intervention on crime deterrence. In particular, selecting experimental and control areas for an experimental design is difficult.

Seventh, in order to determine whether the installation of a CCTV camera has a statistically significant effect on the number of crimes, an appropriate control area should be assigned to each experimental area [33]. Ideally, the residents in these areas should have similar socio-economic characteristics [4], land cover and land use should be the same [3], and the crimes identified before cameras are installed should have a similar typology [3]. If spatial displacement (diffusion) effect is to be measured, then the first buffer around the installed camera is considered as the experimental area. The second concentric buffer is created around the first buffer. This is the area that experiences an increase in crime due to spatial crime displacement from the first buffer area. Finally, the third buffer area represents the control area [4].

There is a growing interest in evidence based studies on crime using the experimental method [34–36]. The randomized experiment is treated as the "gold standard" for evaluation research [7]. This paper contributes to the discussion on evidence-based policy assessment of preventive impact of CCTV on crime. However, since the conditions (see above) to carry out such an assessment may not be ideal, the task needs to be approached pragmatically. In the next chapter we present a method which was designed to analyze the effectiveness of CCTV systems in the city of Poznań. For this study, spatially aggregated crime data posed the biggest challenge. To overcome this challenge we developed an alternative quasi-experimental method of selecting experimental and control areas for the assessment of the effectiveness of CCTV cameras. This issue is currently one of the most important challenges in spatial analysis [37].

## 3. Description of the Method Developed

To investigate the effectiveness of CCTV systems, crime data were collected for a period of ten-years (2005–2014). In the city of Poznań, the CCTV system is comprised of more than 100 cameras and crime data are available. However, in Poznań, similarly to other cities in Poland one of the main problems in crime analysis is the level of spatially aggregated crime data [38]. Although address-level crime data or crime data with exact coordinates are occasionally available for Polish cities [32], the lowest level of aggregation for crime incidents in Polish cities is the area of a street. The issue exists due to procedures of the Polish police. At the same time, camera locations usually have exact addresses, which gives an opportunity to define experimental areas as those streets on which CCTV cameras have been installed.

In order to resolve the issue of a lack of exact crime data locations, a special method was developed to ensure the selection of experimental and control areas in accordance with the requirements of the quasi-experimental method, while also following the guidelines by Welsh and Farrington [16] concerning CCTV systems efficiency. Selection of experimental areas is easier than finding appropriate control areas. Following Welsh and Farrington [16], we adopted the following criteria:

1.  The experimental area must have at least two cameras per 1 km of street length. This criterion is intended to exclude streets barely affected by the intervention.
2.  The number of crime incidents preceding the intervention (installation of a camera) should be no less than 20.
3.  The total number of crime incidents should not be less than 100 in the ten-year period included in the analysis.
4.  Crime incidents data were acquired for the period ranging from 2005 to 2014; we needed at least one year before and two years after the intervention in order to assess its impact. A two-year period was needed because the exact date of the CCTV camera installation was not known. If a camera is installed in December, the impact of this action will appear in the second year. In our study, only streets on which the first camera was installed no earlier than 2006 and no later than 2013 were taken into account.

We made the following assumptions with regard to finding control areas appropriate for each pre-selected experimental area. First, each experimental area can have more than one control area so as to decrease the influence of a multiplicity of factors unevenly impacting the number of crimes across the study area [39]. For example, the distance from the city center may be a significant factor. Another factor may be the number of pubs in the area [17]. In addition, different crime categories would not be affected in the same way [40]. Designating more than one control area gives us the opportunity to decrease the influence of uncontrolled factors on the effect studied and possible biases [2].

The second assumption concerns the number of crimes themselves. A control area should have a similar number of crime incidents per kilometer of a street prior to the installation of CCTV cameras as does the corresponding experimental area. This fulfills the basic criterion of similarity for the control areas.

The third assumption is relevant to the landscape similarity analysis [41,42], which is a method of modeling space and extracting knowledge from patterns detectable in the landscape [43]. It identifies correlations of the patterns with both processes and the functioning of a landscape system [43]. Research shows that criminal activity changes with landscape variability [40]. The height or density of buildings [44] and the abundance of green areas [40,45] are factors considered in the landscape analysis. Socio-demographic metrics would be relevant variables in a study of the effect of a CCTV system. However, they were not available at the very fine resolution that this study required. Moreover, as per literature [16], street crime is the type of crime most likely to be used to analyze the effect of CCTV. Therefore, the landscape of the public space is an important factor.

The impact that land use has on crime has been discussed in the literature and there is some evidence that certain land cover classes included in this research have a relationship to crime. Land cover classes such as railroads (with streetcars, in the case of Poznań) and (low, medium, high) vegetation can serve as a proxy for parks or green areas. As far as railroads are concerned, Poister [46] discovered that robbery, burglary, and auto theft increased with the opening of Metropolitan Rapid Rail stations in Atlanta, GA. A few years later, in a study in Los Angeles, CA, Loukaitou-Sideris et al. [47] found that the influence of light urban rail stations on the crime distribution must be differentiated and is connected with characteristics of stations and their neighborhoods. McCord and Ratcliffe [48] observed that subway stations attract street robberies in Philadelphia, PA. In a study of Boston, MA, Crewe [49] found that urban linear parks and their neighborhoods show slightly lower levels of property crime. In a very recent study in the city of Baton Rouge, LA, Anliu [50] showed that the composition of crime types for all BREC parks in the city was significantly different from areas outside the parks. For example, a higher proportion of "firearm" and "nuisance" crimes and a lower proportion of theft was discovered in parks compared to non-park areas. Additionally, Anliu [50] found BREC parks to serve as a strong detractor of crime with no spatial crime clusters found inside any of the parks. This result was confirmed in a study of the city of Szczecin, Poland, where green areas, among other land-use types, were found to serve as strong deterrents of crime [32]. Whether the other land cover types that are explored in our research have any association to crime remains an open question due to the lack of empirical studies.

In its basic form, the landscape analysis uses diverse landscape metrics calculated on raster land-use/landcover maps. Two commonly used datasets available from the European Environmental Agency are the Corine Land Cover and the Urban Atlas. These datasets provide low resolution, aggregated data with insufficient details of the landscape diversity at the street level. They are not adequate for our study, however. We referred to a method developed by Dąbrowski [51] to create a land-cover map with a ground resolution of 1 m with the following classes of land cover: Low buildings, medium height buildings, high buildings, constructions, dirt roads, paved roads, railroads, bridges, low vegetation (grass), medium vegetation (shrubbery), high vegetation (trees), arable land, anthropogenic land, and water. Dabrowski's method uses GIS tools to transform an available vector topographic database (at a scale of 1:10,000) combined with a LiDAR (Light Detection and Ranging) dataset into a high-resolution, raster land cover map.

To calculate landscape metrics, areas need to be defined for which these metrics will be calculated. For our study these were the basic spatial units to which crime incidents were attributed. We defined them as 40-m buffer areas around the geometric axis of each street. Next, using the FRAGSTATS software [52], we calculated the percentages of the landscape covered by all the land cover classes as one of the most common landscape metrics used in landscape analysis. This metric captures the influence of landscape diversity and abundance of green areas, density of buildings and, since our classes include buildings divided into three height groups, also the height of buildings.

In short, based on the above-mentioned assumptions, the method to select control areas is the following:

1. First, the aggregation to quarterly crime data has been chosen due to rare occurring crimes. We aggregated quarterly crime incident data for each street and subsequently standardized

them by dividing them by the length of the street. Aggregating crime incidents to shorter time frames would have resulted in many streets with few crime incidents or without any, while aggregating to larger time frames would have reduced time series to such short ones that comparing experimental and control areas would have been impossible.

2.  Second, to find control areas with a similar crime density, i.e., similar number of crimes per 1 km of street length, we calculated the Euclidean distance for each experimental area. The Euclidean distance is a good measure for changes of proportions of land-cover types between two time series. It is calculated as:

$$d_{incidents}(p,q) = \sqrt{\sum_{i=1}^{n}(q_i - p_i)^2},$$

where $p = (p_1, p_2, \ldots, p_n)$, $q = (q_1, q_2, \ldots, q_n)$. Values of vectors $p$ and $q$ represent the number of crime incidents per kilometer in each consecutive quarter ($i$) prior to the intervention. For each pair of values $(p_i, q_i)$, small differences are suppressed towards zero and large ones are penalized. The formula is calculated for the experimental area and every street that could become its control area. For example, having a street Q (an experimental area on which a camera has been installed in 2014) and a street P (a possible control area), we have a time series consisting of four quarters before intervention. Supposing the crime density in the experimental area (Q) equals to 10, 9, 8, 7, and the corresponding values for control areas (P) are 8, 5, 3, and 11, the Euclidean distance between the two areas equals to 7.8. The higher the Euclidean distance, the more dissimilar the crime densities are between experimental and control areas, and vice versa.

  Since the distribution of Euclidean distances was strongly right-skewed, we decided to declare a mean distance of seven crimes per 1 km of street length per a quarter of a year as the maximum distance of similarity. In other words, any control area that exceeds this distance is treated as completely dissimilar to its compared experimental area. The remaining distances with values ≤7 were subsequently standardized and inversed. This resulted in a similarity measure where 0 meant that two areas were fully dissimilar (the Euclidean distance was equal to 7 or more), and 1 that their time series were identical (the Euclidean distance was equal to 0). Using this procedure, the distance of 3.5 had a similarity measure of 0.5 and the distance of 6 resulted in a similarity measure of 0.14. This allowed us to select the 10 most similar control areas in terms of crime densities prior to the intervention. The streets, however, could have had similar time series due to sheer coincidence, which could have happened especially for short time series, such as one year. Therefore, experimental and control areas had to be evaluated in terms of their similarity of landscape as well.

3.  We evaluated the similarity of the landscape between the experimental area and its pre-selected control areas. We calculated the Euclidean distance regarding the landscape metrics, i.e.,

$$d_{landscape}(r,s) = \sqrt{\sum_{i=1}^{m}(r_i - s_i)^2},$$

where $r = (r_1, r_2, \ldots, r_m)$, $s = (s_1, s_2, \ldots, s_m)$. Values of vectors $r$ and $s$ represent percentages of each landscape type ($i$) in the experimental ($r$) and the corresponding control area ($s$). As in the case of $d_{incidents}$, for each pair of values $(r_i, s_i)$, small differences are suppressed towards zero and large ones are penalized.

Euclidean distances were standardized and inversed, but this time the dissimilarity threshold was set to 200%, since two areas cannot be more dissimilar from each other than by 200%. For example, one area could be completely covered by water (100% of land cover), whereas the comparison area could be completely covered by forest (100% of land cover). The Euclidean distance is calculated as the sum of these two percentage values.

## 4. Testing the Method in a Case Study

The study area selected for testing the method was the city of Poznań in western Poland which has a population of about 556,000 inhabitants (census 2011). A police dataset of crime incidents was acquired for the period of 2005–2015. During this time, 56,773 crime incidents were reported in four categories which were selected for this study: (a) Battering (street fight); (b) theft from a car; theft of a car, and breaking into a car; (c) damaging a car; and (d) robbery. These crime categories were chosen because they are so-called "street crimes" and are the most likely to be affected by CCTV cameras [16].

Poznań has almost 2300 streets; crimes have been recorded on 1798 of them. Cameras were installed on 210 streets, but only 106 streets had cameras installed between 2006 and 2013 (the period excludes one year prior and two years after the intervention from the time range of all the acquired data). Only 24 of the 106 streets had more than 100 crime incidents during the analyzed period, and 22 of them had more than 20 crime incidents prior to the intervention. After excluding from the analysis streets with less than two cameras per street kilometer, 11 possible experimental areas were identified.

To calculate the similarity between two time series, we aggregated crime incidents quarterly and calculated the similarity for all selected streets before the year the intervention took place. Figure 1 shows time series of crime incidents per street kilometer of the experimental area (Stanislawa Przybyszewskiego street) and one of its preselected control areas (Murawa street). The figure compares the number of crime incidents in both areas to illustrate the logic of calculating their similarity. However, it does not illustrate the effect of CCTV. Both streets are congested with cars and pedestrians. They are also one of the main roads in the city center with access to tram lines.

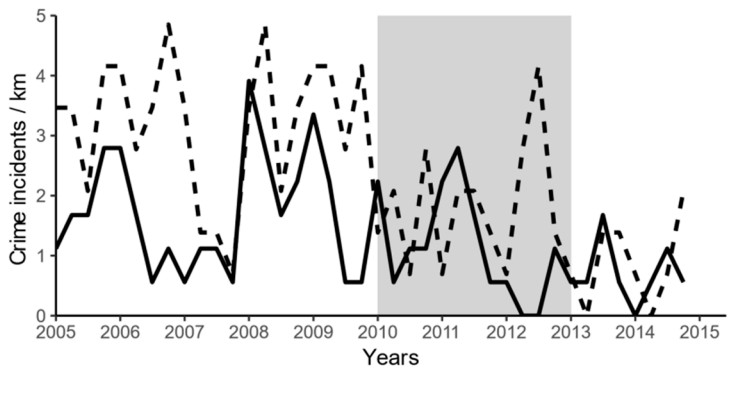

**Figure 1.** The 2005–2015 time series of crime incidents per street kilometer of Stanislawa Przybyszewskiego (experimental area) and Murawa (control area) streets. The grey area denotes the intervention period of 2010–2013.

After selecting the 10 most similar control areas to Stanislawa Przybyszewskiego street, based on the similarity between time series of crime densities, we calculated the similarity between their land cover using percentage of landscape covered by various land cover types—low, medium and high buildings, constructions, dirt roads, paved roads, railroads, bridges, grass, shrub, trees, agricultural area, anthropogenic land (e.g., waste dump or mines) and water. Figure 2 provides an example of land-cover maps for a part of Stanislawa Przybyszewskiego street and one of its control area.

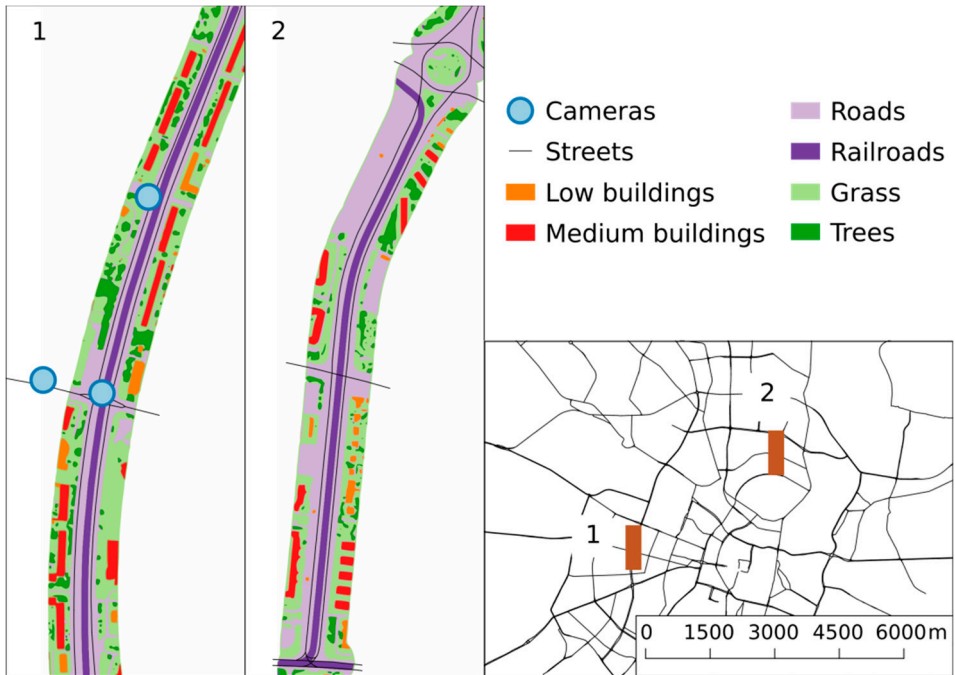

**Figure 2.** Spatial distribution of land cover types for a portion of (**1**) Stanislawa Przybyszewskiego and (**2**) Murawa (**left**) and the location of both street portions in Poznań (**lower right**).

Distances of all 10 preselected control areas and their corresponding experimental area (Stanislawa Przybyszewskiego street) are shown in Figure 3. Similarity between time series of criminal events before the intervention is shown on the vertical axis, while similarity between landscape metrics is shown on the horizontal axis. The larger the values, the more similar the control areas are to the experimental area. From this plot we can select the two most similar streets to the experimental area (Stanislawa Przybyszewskiego street), namely Murawa and Piatkowska streets.

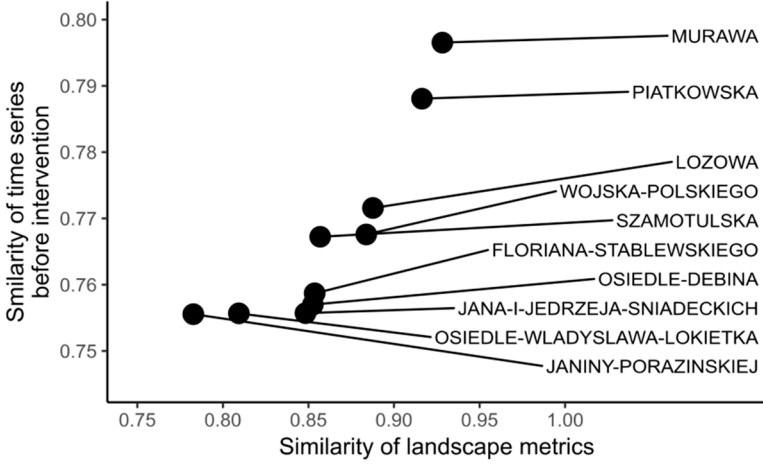

**Figure 3.** The similarity of control areas to experimental area (Stanislawa Przybyszewskiego street).

## 5. Discussion

The method presented in this paper overcomes the difficulty of performing a CCTV effectiveness analysis using the quasi-experimental method when crime incidents are spatially aggregated or are without exact locations. It relies on similarity of crime occurrence time series prior to the intervention and similarity of landscape features between an experimental area and control areas. It must be emphasized that the appropriateness of control areas selection is difficult to evaluate. So far, no objective measures

have been developed that can be used for the best possible selection of control and experimental areas for aggregated crime incident data. However, the advantage of the proposed method over alternative selection methods, i.e., selecting control areas based on subjective criteria and expertise of researchers or experts, is reproducible. It may thus be considered as a step towards finding objective criteria for the evaluation of control and experimental areas selection.

The method developed aims to cope with data weaknesses in crime analysis, especially in the assessment of the effectiveness of CCTV cameras. It is tailored to a case of spatially aggregated data on crime incidents and provides an example of how to deal with such an issue. It was tested in the city of Poznań and appeared feasible. The method was instrumental in selecting experimental and control areas for an analysis in a quasi-experimental study, satisfying quite restrictive, and seldom fully applied, conditions designed by Welsh and Farrington [10]. The method depends on the type and quality of the data, e.g., the lengths of time series, total number of crime incidents, the diversity of the urban fabric in terms of crime density and landscape. Its strength is using similarity metrics and the relationship between landscape and crime incidents. Adjustments to this method may however need to be made in specific cases.

The proposed method is a solution for data weakness, but it is not without limitations. First, it is based on land-cover data at a very fine resolution, which is not always available. Second, we assumed that crime incidents were uniformly distributed along streets, while they may be spatially concentrated along some street segments and absent along other segments. Third, the standardized inversed Euclidean distance was used as a similarity measure for time series proximity (i.e., functions $d_{incidents}(x,y)$ and $d_{landscape}(x,y)$). For each pair of values $(x_i, y_i)$, small differences are suppressed towards zero and large ones are penalized. However, the selection of the distance can influence results. Euclidean distance is a special case of $L_p-norm$ distance, i.e., $L_p(x,y) = \sqrt[p]{\sum_{i=1}^{m}(x_i - y_i)^p}$, where $p = 2$. Further development of our innovative method could focus on evaluating alternative similarity measures, in addition to the distance measure applied in this research. It should not be limited to checking different values for $p$ (e.g., if $p = 1$, then $L_p$ is named the Manhattan distance or city block distance). Research is needed to determine which alternative similarity measures (or possibly their combinations) are suitable for a given problem [53,54]. Furthermore, research is needed to analyze the relationship between landscape and crime incidents. Other landscape metrics, such as the size of patches, or the Shannon diversity index, could explain more suitable relationships to crime incidents. There may thus be better measures to assess landscape similarity than percentage of landscape covered by land-cover classes. Finally, CCTV appears to perform better at reducing some types of crime, such as vehicle theft in parking lots. It may be less effective in reducing emotional crimes (e.g., provocation that leads to violence). These outcomes seem to suggest that quasi-experimental studies should be crime type specific and theory driven.

**Author Contributions:** Conceptualization, A.D., P.M., A.W.; Data curation, A.D., A.W.; Formal analysis, P.M., A.W., M.L.; Funding acquisition, P.M.; Investigation, A.D., P.M.; Methodology, A.D., P.M., A.W., M.L.; Project administration, P.M.; Resources, A.D.; Supervision, P.M.; Visualization, A.D., A.W.; Writing—original draft, A.D., P.M.; Writing—review & editing, A.D., P.M., A.W., M.L.

**Funding:** This paper was prepared within the MoWiz project, no. 2016/21/B/HS6/01158 financed by the National Science Centre.

**Acknowledgments:** Contribution of Jakub Lewandowski at the earlier state of the research is gratefully acknowledged. The paper benefited from comments from the three anonymous reviewers.

**Conflicts of Interest:** The authors declare no conflict of interest.

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
