# Peer review of "Identification of Experimental and Control Areas for CCTV Effectiveness Assessment—The Issue of Spatially Aggregated Data"

_ijgi, doi:10.3390/ijgi7120471_

Round 1
Reviewer 1 Report
Revision 1:
If you are demonstrating this as possible method for quasi-experimental criminology then this is fine but your prerequisites listed on P3 (lines 91-97) are from single source and fairly dated (although definitive reference for CCTV). Please provide additional justification or highlight relevant to CCTV rather than “Identification of Experimental and Control Areas – the Issue of Spatially Aggregated Data in Crime Analysis” as per title.
Revision 2
You highlight mixed evidence regarding CCTV. It is better at reducing crime in controlled environments (eg vehicle theft in car parks) and for less emotional/responsive crimes (eg provocation leading to violence). Therefore again this is fine to use an as example but be explicit – this is an example. I would also stress need for any such quasi-experimental studies to be crime type specific and theory driven – what crime types might that initiative/prevention measure actually reduce.
Revision 3:
This needs theoretical discussion on prevention/detection elements of CCTV. I think focus of paper is on prevention so this needs to be explicit. Note prevention only effective when there is a response to a camera. Some of this is discussed but how does this impact on methodology you propose.
Revision 4:
The work on micro places of crime suggests need to analyse crime at micro places. This evaluation methodology suggests impacts of crime prevention measure can be examined at meso level and I am not convinced this is case. Even if find an effect – how can you put this in context. This should also link to work around realistic evaluations and mechanisms for change.
Revision 5: I am not convinced the land cover metrics you use are appropriate for crime analysis –I would have anticipated some socio-demographic measures that are more relevant.
Revision 6 – Disaggregating at this scale reduced ability to argue against counter-factual. What would have happened without intervention? I worry as disaggregate further how account for this.
Revision 7 – I would have preferred some more discussion of results to give more credence to the findings. At present I am rather sceptical about this. Moe justification and explanation is required, alongside a thorough consideration of limitations
Limitation 8 – At present there is growing concern for evidence –based approaches – especially though randomised control trials. How does your methodology fit within this current academic space.
Author Response
Revision 1:
If you are demonstrating this as possible method for quasi-experimental criminology then this is fine but your prerequisites listed on P3 (lines 91-97) are from single source and fairly dated (although definitive reference for CCTV). Please provide additional justification or highlight relevant to CCTV rather than “Identification of Experimental and Control Areas – the Issue of Spatially Aggregated Data in Crime Analysis” as per title.
Thank you for this suggestion. We modified the title of the paper. It now refers directly to CCTV. Lines 92-96 are rephrased to clarify prerequisites concerning the method to assess CCTV.
Revision 2
You highlight mixed evidence regarding CCTV. It is better at reducing crime in controlled environments (eg vehicle theft in car parks) and for less emotional/responsive crimes (eg provocation leading to violence). Therefore again this is fine to use an as example but be explicit – this is an example. I would also stress need for any such quasi-experimental studies to be crime type specific and theory driven – what crime types might that initiative/prevention measure actually reduce.
Thank you for this comment. Indeed, our proposal is an example and it requires theoretical considerations. We added an explanation concerning these issues in lines 42-44, 49-50 and 75-79. Also we added some text regarding this issue to the discussion (lines 364-367).
Revision 3:
This needs theoretical discussion on prevention/detection elements of CCTV. I think focus of paper is on prevention so this needs to be explicit. Note prevention only effective when there is a response to a camera. Some of this is discussed but how does this impact on methodology you propose.
Thank you for this detailed comment. We added a clarification on our focus on prevention in lines 77-79. We focus on the prevention only, and we declare this. Therefore, we do not discuss the response aspect.
Revision 4:
The work on micro places of crime suggests need to analyse crime at micro places. This evaluation methodology suggests impacts of crime prevention measure can be examined at meso level and I am not convinced this is case. Even if find an effect – how can you put this in context. This should also link to work around realistic evaluations and mechanisms for change.
Thank you for raising this important issue. First of all, our focus on the meso-level is determined by the data availability. It is an external constraint and our intention is to deal with it. Second, although exact places and their characteristics are definitely crucial for crime incidents, the meso-level is also an important determinant. Police aggregates data on crime incidents to the level of specific neighborhoods or districts, since these enumeration units are a combination of unique landscapes and their surroundings that impact crime densities in the city. For example, dense shrubbery along the side of the street can block the view, making the criminal less visible from apartment windows, as well as demographic and economic aspects of the neighborhood, which can be analyzed in a bigger context. We address this issue in the answer to revision 5 .
Revision 5: I am not convinced the land cover metrics you use are appropriate for crime analysis –I would have anticipated some socio-demographic measures that are more relevant.
Thank you for this important comment. We searched the literature and found out that some of our land cover metrics (railroads and low, medium, and high vegetation as a proxy for green areas and parks) had indeed a relationship with crime. We added a paragraph on this in lines 198-220. However, socio-demographic metrics were not available at the fine resolution that our study is based upon. Finally, the crime types that we analyzed are typical “street crime” types. For these, our selected land cover types in the public space are an important factor.
Revision 6 – Disaggregating at this scale reduced ability to argue against counter-factual. What would have happened without intervention? I worry as disaggregate further how account for this.
Thank you for raising the important problem of aggregation. Our work deals with aggregated data. We do not disaggregate data, but adjust them to spatio-temporal frames that allows a comparison between experimental and control areas. In our research a comparison without intervention can only be addressed by using quasi-experimental methods.
Revision 7 – I would have preferred some more discussion of results to give more credence to the findings. At present I am rather sceptical about this. Moe justification and explanation is required, alongside a thorough consideration of limitations
Thank you for this comment. We added more detail in lines 331-337. We came up with a method for selecting control and experimental areas within our constraints of data availability. So far there are no objective measures to the selection of control and experimental areas. However, the advantage of our approach is its reproducibility, compared to alternative selection methods, based on experts’ elicitation, i.e. selecting control areas based on the expertise of experts. We also added more information on limitations of our study, which can be found in lines 348-51.
Limitation 8 – At present there is growing concern for evidence –based approaches – especially though randomised control trials. How does your methodology fit within this current academic space.
Thank you for this relevant comment. We presented this context in our study in lines 45-50. We added lines 49-50 following your comment. We also explain this issues in lines 144-147.

Reviewer 2 Report
The topic of the paper is very interesting.
The authors would like to introduce a novel method to delimit test and control areas in cities to detect the effectiveness of CCTV systems.
My comments.
minor changes are needed in some cases see below:
title- I think this is a bit misleading as you are not dealing with spatially aggregated data but your main concern is to identify test and control areas in case of spatially aggregated data?
line 39 “Although the effect is noticeable for vehicle 38 crimes and car parks, it is usually not significant for city centers and for residential areas”-is this included in Lim? or do you have other references s well?
the formulation of different studies is not clear please define the difference exactly between these: f.e. in line 44 and 55 : quasi-experimental studies ; in line 54 quasi natural experiment method, in line 68 quasi experimental method, in line 83 quasi-experimental scheme, line 150 natural quasi-experiment methodology in line 160 quasi-experimental method (all these are a bit confusing)
also in line 55 you are speaking about randomized experience
I think this part should be more structured and reader-friendly. maybe you can give a short explanation about these methods than explain the problems in your case.
line 101 what do you mean on robust statistical conclusions
in line 103 –what does sufficient mean in this context, please give some exact explanations, numbers
line 103 authors state that experimental/control areas must be as similar as possible –please give bit more explanation about similarity here ( numbers of crime, crime types …)
line 163 –please indicate exactly which literature was involved
in line 171 you mention that we need data 2 years after the interventions. You need to explain this in detailed. I assume that after the installation of the cameras the decrease in the number of crime incidents should be clearly evident.
line 263-265 why these crime types were investigated?-please give some explanation or refer to literature
line 291 figure 2 the colour schemes need to be changed it is very difficult to differentiate medium and high buildings; also the same problem occurs between road and railroad and shrub and trees, the legend even worth as you are using a black outline. Also, it is difficult to see the place of the cameras.
line 294 Maybe another map would be interesting to see the experimental and the preselected 10 control areas
major changes needed in some cases see below:
the authors are continuously speaking about their experiments in 8 Polish cities although giving one example from Poznan, the other 8 should be involved in the paper (line 77)
a more detailed explanation is needed why exactly Euclidean distance method was chosen for the measurement of similarities and dissimilarities
figure 1 the graph is quite contradictory. it seems that in control area crimes per street segments were usually higher always, and after the intervention, we can experience a decline in the crime incidents per street segments also in the control area-
the validation of the given method is completely missing. the authors state that It was tested for a case of the city of Poznan, and appeared feasible. there is no proof for that statement. (line 312)
for the validation and verification of the method they need to develop some criteria or parameters which can be measured with other methods,it can also feasible to ask experts to define experimental and control areas using similarities with their methods and compare the results
Another method can be to use original data (not aggregated) to define experimental and control areas similarities and dissimilarities used in the literature, than spatially aggregate the data after that using the method defined in the paper measure the similarities and dissimilarities and compare the two results.
Author Response
The topic of the paper is very interesting.
The authors would like to introduce a novel method to delimit test and
control areas in cities to detect the effectiveness of CCTV systems.
My comments.
minor changes are needed in some cases see below:
1. title- I think this is a bit misleading as you are not dealing with spatially aggregated data but your main concern is to identify test and control areas in case of spatially aggregated data?
Thank you for this comment. In fact we are dealing with spatially aggregated data, which was the reason for developing this method. We have changed the title to make it coherent with the content of the paper.
2. line 39 “Although the effect is noticeable for vehicle 38 crimes and car parks, it is usually not significant for city centers and for residential areas”-is this included in Lim? or do you have other references s well?
Lim et al. (2016), in their paper provide a review of 11 papers on CCTV impact on crime reduction, so we decided not to repeat/copy this.
3. the formulation of different studies is not clear please define the difference exactly between these: f.e. in line 44 and 55 : quasi-experimental studies ; in line 54 quasi natural experiment method, in line 68 quasi experimental method, in line 83 quasi-experimental scheme, line 150 natural quasi-experiment methodology in line 160 quasi-experimental method (all these are a bit confusing)
Thank you for noticing this inconsistency. Quasi-experiment and quasi-experimental are the core terms, established in the literature. We have gone through the manuscript and made necessary changes to only use these two most appropriate terms.
4. also in line 55 you are speaking about randomized experience
We use the term “randomized experiment” in the sense of Weisburd et al. (2014) as different from the quasi-experimental metod. We addend the reference to Weisburd et al. (2014) on line 50.
5. I think this part should be more structured and reader-friendly. maybe you can give a short explanation about these methods than explain the problems in your case.
Thank you for this comment. We consolidated portions of the randomized experiment description (now in lines 45-50) to better structure this part of the manuscript.
6. line 101 what do you mean on robust statistical conclusions
We rephrased the sentence, to clarify the issue (line 101-2).
7. in line 103 –what does sufficient mean in this context, please give some
exact explanations, numbers
We added explanations referring to statistical tests (line 103).
8. line 103 authors state that experimental/control areas must be as similar as possible –please give bit more explanation about similarity here ( numbers of crime, crime types …)
Thank you for this comment. We clarified the similarity (lines 113-4).
9. line 163 –please indicate exactly which literature was involved
Thank you for this comment. We clarified this sentence by referring directly to Welsh & Farrington (line 168).
10. in line 171 you mention that we need data 2 years after the interventions. You need to explain this in detailed. I assume that after the installation of the cameras the decrease in the number of crime incidents should be clearly evident.
Thank you for rising this issue. We added an explanation about the two year period in lines 177-9.
11. line 263-265 why these crime types were investigated?-please give some
explanation or refer to literature
Thank for noting this important issue. We added a reference to line 206 and explained in lines 290-2, why street crimes were included in the analysis.
12. line 291 figure 2 the colour schemes need to be changed it is very difficult to differentiate medium and high buildings; also the same problem occurs between road and railroad and shrub and trees, the legend even worth as you are using a black outline. Also, it is difficult to
see the place of the cameras.
Colors of Fig. 2 were modified in order to make it clearer.
13. line 294 Maybe another map would be interesting to see the experimental
and the preselected 10 control areas
After considering your suggestion, we think that the additional map would hardly supply new information. Fig. 2 is an example and it presents land cover of streets and the location of these streets on the city map. To us adding further experimental and control areas would be repetitive and not that much informative.
major changes needed in some cases see below:
14. the authors are continuously speaking about their experiments in 8 Polish cities although giving one example from Poznan, the other 8 should be involved in the paper (line 77)
Thank for raising this issue. Indeed, referring to eight cities is unnecessary and confusing, since in this paper we only test the method for the city of Poznan. We removed the reference to eight cities.
15. a more detailed explanation is needed why exactly Euclidean distance method was chosen for the measurement of similarities and dissimilarities
Thank you for this comment. We explained the rationale of using the Euclidean distance in lines 248-50.
16. figure 1 the graph is quite contradictory. it seems that in control area crimes per street segments were usually higher always, and after the intervention, we can experience a decline in the crime incidents per street segments also in the control area-
Thank you for this comment. Fig. 1 is to illustrate the similarity between control and experimental areas. This figure does not show the effectiveness of CCTV. We added an additional explanation in lines 302-04.
17. the validation of the given method is completely missing. the authors state that It was tested for a case of the city of Poznan, and appeared feasible. there is no proof for that statement. (line 312) for the validation and verification of the method they need to develop
some criteria or parameters which can be measured with other methods,
It is a very serious and difficult issue. The validation of the method would be possible although very difficult. However, this was beyond the scope of our study. The feasibility of the method we mention refers to fulfilling the quite restrictive conditions on measuring the CCTV effect, as proposed in the literature [e.g., by Welsh, B. C.; Farrington, D. P. Effects of closed-circuit television on crime. Ann. Am. Acad. Pol. Soc. Sci. 2003, 587, 110–135; Farrington D, Welsh B. Randomized experiments in criminology: what have we learned in the last two decades? Journal of Experimental Criminology, 2005. 1:9–38). In fact these conditions (concerning time series, the number of cases, the number of crimes, etc.) are seldom fulfilled in studies. Our proposed method focuses on the selection of the experimental and control areas. We believe that it advances knowledge in this respect. The validation exercise would be great, but it is a topic for another study.
18. it can also feasible to ask experts to define experimental and control areas using similarities with their methods and compare the results
Thank you for this comment. We deliberatively decided not to rely on experts’ elicitations, as explained in lines 331-7.
19. Another method can be to use original data (not aggregated) to define experimental and control areas similarities and dissimilarities used in the literature, than spatially aggregate the data after that using the method defined in the paper measure the similarities and dissimilarities
Thank you for this suggestion. The initial motivation of our study stemmed from the fact that we could obtain aggregated data only (as explained in lines 155-58). Thus, as far as we know, the alternative approach that you are suggesting is, unfortunately, not possible for us to apply.

Reviewer 3 Report
The authors proposed a method of selecting experimental and control areas for CCTV systems based on time series and landscape analysis in Poznań. In my opinion, the paper has 2 flaws and 3 minor errors:
Major
1- The literature review (section 2) was based on all the implementation limitations of the already known quasi-experimental scheme. Section 3 was just a description of the proposed method. It seems that the new method needs more connexion with the literature.
2- In my point of view, the main problem is that there is a lack of evidence between land cover and crime in an urban context. The studies of geography of crime or environmental criminology are based on land use. Thus, the size of buildings cannot be related to their function. For instance, according to the landscape metrics, you can identify two high buildings in different streets. However, one can be a shopping mall and other can be a residential building. The crime dynamics are going to be different impacting the experiment.
Minor
3- The title (Identification of Experimental and Control Areas – the Issue of Spatially Aggregated Data in Crime Analysis) doesn’t specify the keyword – CCTV. Experimental and control areas of what?
4- The line spacing is different in section 1 if comparing with others sections.
5- The paragraph which starts in line 320 (page 10) seems out of context in the discussion.
Author Response
The authors proposed a method of selecting experimental and control areas for CCTV systems based on time series and landscape analysis in Poznań. In my opinion, the paper has 2 flaws and 3 minor errors:
Major
1- The literature review (section 2) was based on all the implementation limitations of the already known quasi-experimental scheme. Section 3 was just a description of the proposed method. It seems that the new method needs more connexion with the literature.
Thank you for this important comment. We referred to the literature on evidence based assessments (lines 144-147), linking our study to methodological considerations related to the experimental method and crime policy assessment.
2- In my point of view, the main problem is that there is a lack of evidence between land cover and crime in an urban context. The studies of geography of crime or environmental criminology are based on land use. Thus, the size of buildings cannot be related to their function. For instance, according to the landscape metrics, you can identify two high buildings in different streets. However, one can be a shopping mall and other can be a residential building. The crime dynamics are going to be different impacting the experiment.
Thank you for raising this issue. Indeed, we base our research on urban ecology studies that connect the structure of the landscape to its functions and the processes that occur within the structure. Although, it is true that one specific building can have different functionalities, when we look from bigger perspective it is usually intuitively easy to delimit the city center, or industrial or commercial areas based on satellite imagery or land cover maps. Since we are working with aggregated data, such land cover data give us more in-depth information of the underlying structure, without losing much of the concept of its land use (see lines 202-20).
We also searched the literature and found out that some of our land cover metrics (railroads and low, medium, and high vegetation as a proxy for green areas and parks) had indeed a relationship with crime. We added a paragraph on this in lines 198-201.
Minor
3- The title (Identification of Experimental and Control Areas – the Issue of Spatially Aggregated Data in Crime Analysis) doesn’t specify the keyword – CCTV. Experimental and control areas of what?
Thank you for your suggestion. The title was changed in accordance with your consideration.
4- The line spacing is different in section 1 if comparing with others sections.
Thank you. It was corrected.
5- The paragraph which starts in line 320 (page 10) seems out of context in the discussion.
We added a sentence linking the paragraph with the preceding storyline on limitations of our study (line 354-5).

Round 2
Reviewer 1 Report
Figure 1.
I can' tell difference between streets. This needs a legend that distinguishes between these.
Author Response
Comment1: I can' tell difference between streets. This needs a legend that distinguishes between these.
Thank you for noticing this deficiency. We corrected it and now the streets can be distinguished.
Reviewer 3 Report
The authors improved the original version significantly. Then, I accept the paper in present form.
Author Response
The authors improved the original version significantly. Then, I accept the paper in present form.
Thank you very much for your constructive comments and for accepting our corrections.